# An Evidence-Based Somatic Acupressure Intervention Protocol for Managing the Breast Cancer Fatigue-Sleep Disturbance-Depression Symptom Cluster: Development and Validation following the Medical Research Council Framework

**DOI:** 10.3390/ijerph191911934

**Published:** 2022-09-21

**Authors:** Jing-Yu (Benjamin) Tan, Tao Wang, Isabella Zhao, Mary Janice Polotan, Sabina Eliseeva

**Affiliations:** 1College of Nursing and Midwifery, Charles Darwin University, Casuarina, NT 0810, Australia; 2College of Nursing and Midwifery, Charles Darwin University, Brisbane Centre, 410 Ann Street, Brisbane, QLD 4000, Australia; 3Cancer and Palliative Care Outcomes Centre, Queensland University of Technology, Kelvin Grove, Brisbane, QLD 4000, Australia; 4Thornlands General Practice, Thornlands, QLD 4164, Australia

**Keywords:** breast cancer, fatigue, sleep disturbance, depression, acupressure, content validity

## Abstract

Background: Somatic acupoint stimulation (SAS) has been frequently utilised as a promising intervention for individual cancer-related symptom management, such as fatigue, sleep disturbance and depression. However, research evidence regarding the role of SAS in mitigating the fatigue-sleep disturbance-depression symptom cluster (FSDSC) has been scant. This study was conducted to develop an evidence-based SAS intervention protocol that can be further implemented in a Phase II randomized controlled trial (RCT) to manage the FSDSC in breast cancer survivors. Methods: The Medical Research Council Framework for Developing and Evaluating Complex Intervention (MRC framework) was employed to guide the development procedures of the SAS intervention protocol, including the identification of an existing evidence base, the identification of theories and practice standards, and the validation of the SAS intervention protocol. A content validity study was performed through an expert panel to assess the scientific and practical appropriateness of the SAS intervention protocol. The content validity index (CVI), including item-level CVI and protocol-level CVI, were calculated to evaluate the consensus level of the expert panel. Results: Key components of the SAS protocol, including the acupoint formula, the SAS modality, technique, intensity and frequency were identified for both a true and placebo SAS intervention based on the best available research evidence retrieved from systematic reviews, clinical trials, and relevant theories, particularly regarding the inflammatory process, *yin-yang* theory, *zang-fu* organs and meridians theory, and acupressure practical standards. The true SAS intervention was determined as daily self-administered acupressure on specific acupoints for seven weeks. The placebo SAS was designed as light acupressure on non-acupoints with the same frequency and duration as the true SAS. Excellent content validity was achieved after one round of expert panel assessment, with all the key components of the true and placebo SAS protocols rated as content valid (CVI ranged from 0.86 to 1.00). Conclusions: A research-informed, theory-driven and practically feasible SAS intervention protocol for the FSDSC management in breast cancer survivors was developed following the MRC framework. The feasibility and acceptability of the SAS intervention will be further tested in breast cancer survivors through a Phase II RCT.

## 1. Background

Fatigue is one of the most common symptoms in breast cancer survivors (BCS) throughout their illness trajectory [1]. Around half of BCS experience significant fatigue before the anticancer treatment, and the prevalence has been reported to be even higher during the posttreatment period, with the incidence ranging from 64% to 75% [2,3,4]. Sleep disturbance and depression are also two frequently reported symptoms among BCS at the posttreatment stage, with incidences of 47% to 65% and 25% to 39%, respectively [2,5,6,7]. Fatigue, sleep disturbance and depression usually do not present as isolated symptoms, but frequently co-occur as a symptom cluster [8] which is defined as “three or more concurrent symptoms that are related to each other” (p. 465) [9]. Fatigue, sleep disturbance and depression are interrelated, and fatigue has been generally regarded as the core symptom since it can predict the other two symptoms [8,10]. The fatigue-sleep disturbance-depression symptom cluster (FSDSC) can negatively impact cancer survivors’ well-being and functional status, deteriorate their quality of life and immunity, lead to various complications, and can increase financial burdens as well as healthcare utilisation, all of which can subsequently reduce cancer survivors’ compliance to treatment, and can even decrease long-term survival rates [8,11,12,13,14,15].

Due to the complex mechanisms of the cancer symptom cluster, no tailored pharmaceutical agents have been proposed as yet for FSDSC management [13]. Contradictory treatment effects have been found for many pharmaceutical agents, such as melatonin and paroxetine [13,16,17]. Meanwhile, drug-related adverse effects and the potential drug interactions with concurrent anticancer regimens [14,16,18,19] also limit the level of evidence and the recommendation of pharmacological interventions for FSDSC management. Excessive use of pharmaceutical agents may also further worsen cancer survivors’ chemical [20] and financial burdens. Therefore, there is a strong need to explore the role of non-pharmacological methods for cancer symptom cluster management as adjuvant approaches to pharmaceutical agents.

Several non-pharmacological interventions, such as cognitive behavioral therapy (CBT), physical exercise, acupuncture and yoga have been used for cancer symptom management with demonstrated beneficial effects [1,13,20,21]. However, those non-pharmacological interventions were mainly focused on managing individual symptoms, not the entire symptom cluster. Feasibility issues of some non-pharmacological approaches have also been a concern. For instance, CBT and yoga usually require extensive professional support, and survivors may need to travel frequently to particular settings for intervention, which can be considerably time- and energy-consuming. Professionally supervised interventions also limit the space for long-term home-based self-management of cancer symptoms. Physical exercise may induce a potential recruitment bias, since fatigue can hinder study participation in research involving exercise training [11]. Considering the long-term experience of FSDSC in BCS, other home-based non-pharmacological interventions that can be self-managed by survivors are worthy of further exploration.

Somatic acupoint stimulation (SAS), including body acupuncture and acupressure, has been frequently utilised to manage various health conditions such as pain, insomnia, and fatigue. Existing evidence supports a promising role for SAS in the individual symptom management of cancer-related fatigue, sleep disturbance and depression [19,22,23,24,25,26], although robust evidence has been inconclusive due to unsatisfactory methodological quality and the limited sample size identified in some studies. Meanwhile, the intervention protocols of SAS, including acupoint formula, stimulation technique, frequency, and duration have also varied significantly across studies [22,25]. A SAS protocol with evidence-based optimal intervention dosage and acupoint formula has been lacking, and no studies on SAS have ever been conducted for FSDSC management. This study was therefore conducted with the aim of developing and validating an evidence-based SAS intervention for FSDSC management in BCS.

## 2. Methods

The abstract of this manuscript has been accepted and presented in the Multinational Association for Supportive Care in Cancer (MASCC)/International Society for Oral Oncology (ISOO) 2022 Annual Meeting, Toronto, Canada. A preprint version of this paper is also available via: https://www.researchsquare.com/article/rs-1821735/v1 (accessed on 10 August 2022).

### 2.1. Study Design

The study procedures follow the Medical Research Council Framework for Developing and Evaluating Complex Intervention (MRC framework), which outlines a “development–evaluation–implementation” framework for establishing an evidence-based complex intervention with four design phases [27]: (1) phase I: developing the intervention protocol; (2) phase II: examining the intervention feasibility and piloting its methodological procedure; (3) phase III: evaluating the interventional effects; and (4) phase IV: utilising and disseminating the intervention. This study followed phase I of the MRC framework to develop and validate an evidence-based SAS intervention protocol which included three major components: (1) identification of existing research evidence; (2) identification of relevant theories and practice standards; and (3) validation of the intervention protocol. This study is the phase I study of a large research project examining the value of acupressure for symptom management in early-stage BCS. Ethical approval of the project was granted by the Human Research Ethics Committee at Charles Darwin University in Australia (H19017). Procedures of the current study are presented in Figure 1.

### 2.2. Study Procedures in Developing and Validating the SAS Intervention Protocol

#### 2.2.1. Identification of Existing Research Evidence Base

The MRC framework suggests that an appropriate intervention should be systematically developed and based on reliable evidence that is identified in research literature and relevant theories [27]. In this study, findings from six systematic reviews and three clinical trials [28,29,30] were adopted to identify the relevant research evidence base. Of the six systematic reviews, three (two were published [31,32] and the other one was under review) on SAS for FSDSC management in cancer patients (each review focused on one single symptom within the FSDSC) were conducted by our research team, while another three systematic reviews on the safety issues of acupressure [33] and placebo acupoint stimulation design [34,35] were utilised to inform the selection of an appropriate SAS modality and the development of a placebo SAS group, respectively.

#### 2.2.2. Identification of Relevant Theories and Practice Standards

The MRC framework emphases the importance of utilising appropriate theories to guide the design of an intervention by clarifying the potential mechanism regarding the effectiveness of the proposed intervention [36,37]. A comprehensive literature search was therefore conducted to locate appropriate theories to support the use of SAS for cancer symptom management. Databases including PubMed, Google Scholar, and China National Knowledge Infrastructure (CNKI) were searched using “acupressure”, “acupuncture”, “acupoints stimulation”, “theory”, and “fatigue”, in combination with a manual search of books on Traditional Chinese Medicine through the university library and a search of reference lists of reviewed articles. Neurophysiological theories and some TCM theories, particularly the inflammatory theory, *yin-yang* theory, and *zang-fu* organs and meridians theory, were identified and utilised to theoretically clarify the potential mechanisms regarding FSDSC in BCS and to inform the selection of an appropriate intervention. Practice standards and handbooks on acupoint stimulation [38,39] were also used to provide rationale regarding the selection of acupoints, modality, intensity and technique, and dosage for the SAS treatment.

#### 2.2.3. Validation of the SAS Intervention Protocol

The MRC Framework suggests that the context in which the intervention will take place should be considered when developing an intervention [27]. A content validity study was conducted through a panel of experts to assess the scientific and practical appropriateness of the SAS intervention protocol. The panel consensus was determined by using the content validity index (CVI) [40]. A panel of seven experts was established. According to Lynn [40], a satisfactory CVI for each assessment item within an intervention protocol must be at least 0.86 when there are seven experts, which indicated that at least six experts in the panel should agree that the item is content valid [40]. Experts were those who met the following inclusion criteria: (1) were registered health practitioners or academics; (2) had more than 10 years of research and/or practice experience in the field of oncology, TCM and/or acupoints stimulation; (3) held a senior academic or clinical position (associate physician, associate professor or above); and (4) were willing to be involved in the content validation study.

All panel experts were recruited through a purposive sampling approach. The form of the content validity assessment included two parts: the true SAS protocol and the placebo SAS protocol. Each protocol had five items for assessment, namely “the acupoints formula”, “the SAS modality”, “the SAS intensity and technique”, “the SAS frequency and session”, and “the SAS total duration”. Each item was assessed using a 4-point Likert scale (from 1= “totally inappropriate” to 4= “very appropriate”). Comments and suggestions were requested from the panel when the item was rated as “totally inappropriate” or “inappropriate” to assist with the further refinement of the protocol.

Content validity of each item (Item level CVI, I-CVI) and entire protocol (scale-level CVI, S-CVI) were assessed [40,41]. Given that a satisfactory I-CVI should be no less than 0.86, at least six out of the seven experts in this study should rate the item as “appropriate” or “very appropriate” [40]. The S-CVI was the proportion of items that achieved a satisfactory I-CVI, and a satisfactory S-CVI should be at least 0.9 [42]. Items with an unsatisfactory CVI should be further revised until satisfactory CVI values were achieved.

## 3. Results

### 3.1. True SAS Intervention Protocol

The true SAS intervention protocol was developed through the following four procedures: (1) selecting an appropriate SAS intervention and acupoints formula; (2) selecting an appropriate SAS modality; (3) identifying an appropriate SAS intensity and technique; and (4) identifying the appropriate SAS dosage. Table 1 summarises the contents of the true SAS intervention protocol, as well as relevant justifications and evidence sources.

#### 3.1.1. Selecting Appropriate SAS Intervention and Acupoints Formula

Neurophysiological theories, particularly the inflammatory theory, and some TCM theories, including the *yin-yang* theory and the *zang-fu* organs and meridians theory, were used to describe the potential mechanism of SAS for FSDSC management in BCS. Inflammation has been regarded as the key biological pathway of FSDSC, since inflammatory cytokines can result in fatigue (the core symptom of the FSDSC) via the autonomic nervous system and/or the hypothalamic–pituitary–adrenal axis [11,43]. An increase in inflammatory markers can also affect immune regulation and recovery via the cellular immune system [11]. Current evidence has demonstrated an important role of acupoint stimulation in decreasing inflammatory cytokines and modulating immune functions [44]. Such potential biological functions of acupoint stimulation involving the regulation of inflammatory cytokines highlight its promising role in FSDSC management given that the development of cancer symptoms is believed to be closely linked with the inflammatory responses induced by inflammatory cytokines [2].

TCM believes that the main pathogeneses of FSDSC are the imbalance between *yin* and *yang*, and deficiencies of *qi* [45,46]. The imbalance between *yin* and *yang* can negatively affect the operation of *qi* and meridians, and subsequently distort the normal *zang-fu* organ functions, given that each major meridian is closely associated with specific *zang-fu* organs (heart, liver, spleen, lungs and kidney). *Qi* refers to the vital energy of the body, which can maintain blood circulation, a warm body, and fights diseases [47]. *Qi* deficiency can significantly contribute to a range of deficiency syndromes in cancer patients, including fatigue [47]. Acupoints are the areas where the *qi* of *zang-fu* organs and meridians are transfused [44]. According to the *yin-yang* theory and the *zang-fu* organs and meridians theory, stimulating specific acupoints can promote the flow of *qi of the body* to maintain blood circulation, dredge the meridians, balance *yin* and *yang*, and regulate *zang-fu* functions [44,48,49].

In accordance with the TCM theories and the relevant acupoint stimulation practice standards/handbooks regarding the indications, effects and roles of somatic acupoints [38,39], eleven acupoints were selected for use in the intervention protocol, including Zusanli (ST36), Sanyinjiao (SP6), Taixi (KI3), Hegu (LI4), Neiguan (PC6), Shenmen (HT7), Baihui (GV20), Qihai (CV6), Guanyuan (CV4), Yintang (EX-HN3), and Taichong (LR3). The selection of these acupoints was also supported by our systematic reviews as the commonly utilised acupoints for alleviating the symptoms of cancer-related fatigue, sleep distress and depression, with demonstrated benefits for symptom improvement (details see Table 1). Two practice standards/handbooks of acupoint stimulation [38,39] were used to inform the accurate location of each acupoint included in the protocol. Details of the indications, effects, roles, and locations of each selected acupoint are listed in the Appendix A.

#### 3.1.2. Selecting Appropriate SAS Modality

Considering that the SAS has two modalities—somatic acupuncture and somatic acupressure, evidence from systematic review findings and clinical trials [28,29,30,31,32] was adopted to support the selection of an appropriate SAS modality. Due to the popularity of somatic acupressure for symptom management in cancer populations and its superiority over invasive somatic acupuncture regarding the risk–benefit balance, convenience, and safety [28,29,30,33], the noninvasive somatic acupressure was identified as an appropriate SAS modality. In comparison with somatic acupuncture, which is administered by qualified practitioners and is an invasive method with potentially severe adverse effects, the noninvasive somatic acupressure method can be a safe method for the long-term self-management of cancer symptoms.

#### 3.1.3. Identifying Appropriate SAS Intensity and Technique

*Deqi* sensation, as a key indicator for the achievement of satisfactory treatment effects of acupoint stimulation, has been widely utilised in research and practice [31,50]. *Deqi* usually refers to a local sensation of dull pain, aching, sore, numb, distended, heavy, electric, throbbing, and warmness [50]. In this study, the acupoints stimulation intensity was determined using the indicator of *deqi* sensation. For the self-acupressure technique, practice standards of finger acupressure were adopted, which include the skills of pointing, pressing, and kneading. Pointing is the patient locating the acupoint [28,51]. Pressing is the patient using either thumb or index finger to press the point to evoke the *deqi* sensation [51]. Kneading is the patient rotating their thumb or index finger on the identified acupoint to achieve therapeutic effects [51].

#### 3.1.4. Identifying Appropriate SAS Dosage

The SAS dosage, including frequency, session and total duration, was determined based on the standards of acupressure [52], and research evidence from systematic reviews and high-quality clinical trials [28,31]. The frequency of SAS treatment was commonly scheduled as daily, with acupressure being performed on each acupoint for two to three minutes [52]. Considering that eleven acupoints were selected in this study, seven of which were bilateral acupoints, it was determined that each acupoint should be pressed for two minutes to minimise the burden on participants. For the total duration of SAS, clinical research has well-documented that a minimum of four-week daily acupressure is required to achieve satisfactory effects, and seven-week daily acupressure is required to achieve maximum effects on cancer-related fatigue alleviation (the core symptom of FSDSC) [28]. Considering all the evidence mentioned above, the SAS dosage in this study was scheduled as daily acupressure for seven weeks, with each acupoint being pressed for two minutes.

### 3.2. Protocol of the Placebo SAS Intervention

The placebo SAS intervention protocol was developed based on the evidence extracted from two comprehensive systematic reviews on placebo acupoint stimulation design [34,35]. To promote the success of a satisfactory blind design in future RCT and to minimise the potential treatment effects of the placebo acupressure, non-acupoints located one to three cm away from the active acupoints utilised in the true intervention, but away from the meridians [34], were selected as the placebo acupressure points. The number of non-acupoints was the same as for the true SAS intervention.

#### 3.2.1. Identifying Appropriate Placebo Acupoints

A few placebo acupoint designs are commonly utilised in clinical research, which include non-acupoints, true acupoints (irrelated to the targeted health problem), and the same acupoints as used in the true group, but with light or no stimulation [34,35]. Given that our future RCT will implement a blind design of participants between the true and placebo acupressure groups, true acupoints with light or no stimulation might not be an appropriate design to achieve a successful blind design, particularly for participants with previous experience of acupressure [34]. Meanwhile, research evidence has indicated that even light acupressure at the true points can evoke some therapeutic effects for symptom alleviation such as pain [53]. Irrelevant acupoints (to the targeted health condition) are another commonly adopted placebo acupoint stimulation approach. However, according to the “holism concept” of TCM, stimulation of any acupoint could lead to a comprehensive response of the human body and subsequently create some treatment effects, particularly when participants in true and placebo groups receiving the same acupressure intensity [34,54]. Non-acupoints are defined as inactive points that are located near the true acupoints, but away from the meridians [34,35]. Non-acupoints have been the most commonly utilised placebo approach in acupoint stimulation studies, since this method would theoretically avoid evoking any specific therapeutic effects generated from true acupoint stimulation. Non-acupoints located near true acupoints are also considered as an appropriate design for maintaining a successful blind design [34]. Given together, the non-acupoints approach was identified as the placebo SAS method in this study. The number of non-acupoints was the same as in the true intervention. All the selected non-acupoints were located one to three cm away from the active acupoints used in the true intervention, but away from the meridians [34].

#### 3.2.2. Identifying Appropriate Placebo SAS Intensity, Technique, Frequency, Session and Total Duration

Using an acupressure device has been recommended as an appropriate method to achieve satisfactory blind design, particularly when the devices in the true and placebo group are identical [34]. However, many acupressure devices, such as acupressure band, are only applicable for acupoints located on arms and legs. Given that some of the selected acupoints in this study are on the head and abdomen, it was unrealistic to use an acupressure device on such acupoints. Finger acupressure on the non-acupoints was therefore selected for the placebo intervention in this study.

Placebo acupressure using the same intensity as in the true acupressure intervention may secure a successful blind design. However, there are over 2000 extra points currently identified in the human body which are not linked to the meridians [54]. Therefore, it might be possible that some so-called non-acupoints are potentially active acupoints with specific treatment effects [54]. To minimise the potential therapeutic effects of placebo SAS intervention to the greatest extent, light acupressure on non-acupoints without evoking the *deqi* sensation was utilised in this study. Frequency, session and total duration of the placebo intervention were scheduled the same as for the true intervention (daily placebo acupressure for seven weeks, with each acupoint being lightly pressed for two minutes).

### 3.3. Content Validity of the SAS Intervention Protocol

Seven experts in the field of oncology, TCM and/or acupoint stimulation were recruited to evaluate the content validity of the proposed SAS intervention protocol. The basic information of the expert panel is shown in Table 2. Satisfactory I-CVI and S-CVI were achieved after one round of assessment. All of the five items presented in the true and placebo SAS protocol were determined to be content valid (I-CVI ranged from 0.86 to 1.00 and S-CVI was 1.00). Details of the content validity assessment are presented in Table 3.

## 4. Discussion

This study demonstrated how an evidence-based SAS intervention protocol for FSDSC management in BCS was developed and validated systematically following the MRC framework. In order to develop a research-informed, theory-driven, and practically feasible SAS intervention, an evidenced-based approach was adopted in this study with multiple supporting evidence from systematic reviews and clinical trials (the research evidence base), relevant modern medicine and TCM theories, SAS practical standards (knowledge and practice evidence base), and a content validity study (expert panel consensus). The methodological procedures presented in this study can provide important implications for future studies in the fields of integrative oncology and complementary therapies to develop evidence-based complex healthcare interventions that are tailored to specific health conditions.

Utilising appropriate methodological guidance can enhance the quality of research design, the likelihood of success of an intervention, and reduce ‘research waste’ [55,56]. In this study, the MRC framework, one of the most commonly used guides for complex intervention development and evaluation, was utilised. Following the MRC framework, a thorough identification of existing research evidence on SAS for FSDSC management was conducted prior to the intervention development, to locate the best available research evidence base. Considering that high-quality systematic reviews are viewed as ideal and reliable sources of research evidence [56], findings from six well-conducted systematic reviews were utilised to build the research evidence base of this study for the SAS intervention development. Learning the theories that underpin the proposed intervention can enhance the understanding of the mechanisms and actions within the causal chain and reduce the risk of developing a non-effective intervention [56]. In this study, relevant theories, including the inflammatory theory, the *yin-yang* theory, and the *zang-fu* organs and meridians theory were adopted to clarify the causal mechanisms in using SAS for the alleviation of FSDSC in BCS. Systematic reviews provided key evidence regarding the identification of the most commonly used acupoint formula, SAS modality and dosage for the management of cancer-related fatigue, sleep disturbance and depression, with strong theoretical and practical rationale supported by relevant theories and practice standards. Developing the intervention through a combination of the best available research evidence, theories and practice standards is a distinguished feature of this study to guarantee that the study procedures are research-informed, theory-driven and practically appropriate.

A “placebo control” is commonly designed in acupoint stimulation studies to distinguish the specific effects of acupoint stimulation from its non-specific effects (placebo effects) [35]. A placebo SAS method was developed in this study for use in future clinical trial to help identify and size the specific effects of SAS for FSDSC management. Given that an inappropriate placebo method might result in some unwanted specific effects of acupoint stimulation and might subsequently lead to a misjudgment of the causality between the intervention and outcomes [34], research evidence from large systematic reviews on placebo acupressure design was used in this study to guide the placebo intervention development. Non-acupoints located near the active acupoints used in the true SAS were identified as the placebo SAS design, which could potentially facilitate a satisfactory blind design between the true and placebo interventions. Given that light stimulation of the non-acupoints may let some study participants (particularly those with previous experience of acupressure) identify the differences between the true and placebo interventions (as light stimulation cannot generate any “Deqi” sensation such as tingling, numbness, heaviness), the future clinical trial phase could consider only recruiting SAS-naïve BCS to minimise the risk of breaking the binding design among participants in the placebo group.

Using a consensus method, for instance, involving experienced researchers and clinicians in the process of identifying and standardising the essential components of an intervention protocol, can contribute to a more robust clinical integrity and acceptance of the intervention [57]. In this study, a panel of experts consisting of researchers and clinicians were involved in assessing the proposed intervention protocol. Content validity of the true and placebo SAS interventions was determined as excellent, indicating that the SAS intervention protocol is theoretically and practically appropriate for FSDSC management in BCS. Although an excellent agreement was obtained from the expert panel, suggestions were received to refine the SAS protocol further. One expert suggested that the sequence of acupoints should be specified to be more easily remembered by the study participants and to ensure that all the acupoints are pressed without missing any. The research team, therefore, proposed the following sequence (from head to toe) for both the true and placebo interventions, in accordance with the clinical acupuncturist’s suggestion: (1) Baihui → (2) Yintang → (3) Neiguan → (4) Shenmen → (5) Hegu → (6) Zusanli → (7) Sanyinjiao → (8) Taixi → (9) Taichong → (10) Qihai → (11) Guanyuan.

Some limitations exist in this study. Although an in-depth understanding of the needs, preferences and opinions of lay experts (BCS in this study) is important for intervention development [56], cancer survivors were not included in the content validity study at this stage. Representativity of the expert panel might be another concern as the purposive sampling method was used in the panel recruitment. The SAS protocol developed in this study can only be viewed as a preliminary stage of work, and future study phases, including a Phase II RCT, qualitative interviews, and a fully-powered Phase III RCT, are necessary to further identify the feasibility, acceptability, and clinical utility of the SAS for the management of FSDSC in BCS.

## 5. Conclusions

This study detailed the methodological procedures in developing and validating an evidence-based SAS intervention for the FSDSC management in BCS by following the phase I of the MRC framework and comprehensively adopting the best available evidence from systematic reviews, clinical trials, relevant theories, practical standards, and an expert panel consensus to ensure that the SAS intervention is research-informed, theory-driven, and practically feasible. The feasibility, acceptability and clinical utility of the SAS intervention protocol for the FSDSC management in BCS will be further examined in a Phase II RCT.

## Figures and Tables

**Figure 1 ijerph-19-11934-f001:**
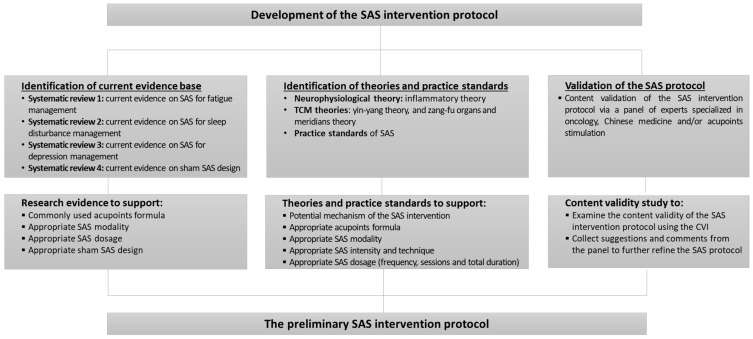
Overview of the study process. SAS: Somatic acupoints stimulation; CVI: Content validity index.

**Table 1 ijerph-19-11934-t001:** Contents of the true SAS intervention protocol with relevant justifications and evidence.

Contents	Details	Justifications	Sources of Evidence
Acupoints formula	▪ Zusanli (ST36) Sanyinjiao (SP6) Taixi (KI3) Hegu (LI4) Neiguan (PC6) Shenmen (HT7) Baihui (GV20) Qihai (CV6) Guanyuan (CV4) Yintang (EX-HN3) Taichong (LR3)	▪ Demonstrated benefits in alleviating cancer-related symptoms of fatigue, sleep distress and depression Commonly utilised acupoints for alleviating cancer-related fatigue, sleep distress and depression in both clinical practice and research	▪ TCM theories: *yin-yang* theory, and *zang-fu* organs and meridians theory [44,48,49] SAS practical standards: instructions of function and clinical usage of somatic acupoints [38,39] Systematic review evidence [31,32] *
SAS modality	▪ Somatic acupressure	▪ Commonly used SAS approach for cancer symptom management Superiority to invasive acupuncture regarding safety and convenience in facilitating self-management in the long run	▪ Systematic review evidence [31,32,33] * Clinical research evidence [28,29,30]
SAS intensity and technique	▪ Intensity: achievement of “Deqi” sensation Technique: Survivors self-administrated manual acupressure using finger pressing	▪ “Deqi” sensation: a key indicator of the achievement of satisfactory therapeutic effects of SAS Acupressure via finger: a common approach of self-acupressure with demonstrated safety and convenience for self-practice	▪ Systematic review evidence [31,32] * SAS practical standards [28,50,51]
SAS frequency, sessions and total duration	▪ A daily SAS for seven weeks, with each session lasting 36 min (2 min on each acupoint)	▪ Commonly used SAS dosage in both research and practice with demonstrated benefits for individual symptom alleviation, appropriate feasibility and acceptability	▪ Systematic review evidence [31,32] * SAS practical standards [52] Clinical research evidence [28]

SAS: Somatic acupoint stimulation; * The systematic review evidence was adopted from three systematic reviews that were conducted by our research team, with two published and one under review (Wang, T., Tan, J.Y., Yao, L.Q., et al. Effects of somatic acupoint stimulation on anxiety and depression in cancer patients: an updated quantitative synthesis of randomized controlled trials. Manuscript submitted for publication). Only the two published systematic reviews [31,32] were included in the reference list.

**Table 2 ijerph-19-11934-t002:** Basic information of the expert panel (N = 7).

Basic Information of the Expert Panel	Percentage (%)
**Working institution**	
University	3 (42.9%)
Hospital	4 (57.1%)
**Age**	
30–40 years old	3 (42.9%)
40–50 years old	2 (28.5%)
50–60 years old	2 (28.5%)
**Highest academic degree**	
PhD or MD	4 (57.1%)
Master’s degree	2 (28.5%)
Bachelor’s degree	1 (14.3%)
**Academic and professional title**	
Full professor	3 (42.9%)
Associate chief physician or associate consultant	4 (57.1%)
**Years of professional experience**	
5–10 years	1 (14.3%)
11–15 years	2 (28.5%)
16–20 years	1 (14.3%)
Over 20 years	3 (42.9%)

**Table 3 ijerph-19-11934-t003:** Content validity assessment of the true and placebo SAS protocol (number of experts = 7).

Item	Description of Each Item	Content Validity Assessment			CVI
		No. of Experts Rating as “(4) Very Appropriate”	No. of Experts Rating as “(3) Appropriate”	No. of Experts Rating as (3) or (4)	
**True SAS intervention protocol**					
1	Selection of acupoints formula	5	2	7	I-CVI 1.00
2	SAS modality	5	2	7	I-CVI 1.00
3	SAS intensity and technique	4	3	7	I-CVI 1.00
4	SAS frequency and sessions	3	4	7	I-CVI 1.00
5	SAS total duration	4	2	6	I-CVI 0.86
**Scale level CVI**					**S-CVI 1.00**
**Placebo SAS intervention protocol**					
1	Selection of acupoints formula	5	1	6	I-CVI 0.86
2	SAS modality	6	1	7	I-CVI 1.00
3	SAS intensity and technique	6	1	7	I-CVI 1.00
4	SAS frequency and sessions	3	4	7	I-CVI 1.00
5	SAS total duration	4	2	6	I-CVI 0.86
**Scale level CVI**					**S-CVI 1.00**

SAS: somatic acupoints stimulation; CVI: content validity index.

## Data Availability

Data are available upon special request.

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
