# Peer review of "An Evidence-Based Somatic Acupressure Intervention Protocol for Managing the Breast Cancer Fatigue-Sleep Disturbance-Depression Symptom Cluster: Development and Validation following the Medical Research Council Framework"

_ijerph, 2022, doi:10.3390/ijerph191911934_

Round 1

Reviewer 1 Report

The article entitled “An evidence-based somatic acupressure intervention for managing the breast cancer fatigue-sleep disturbance-depression symptom cluster: Development and validation following the Medical Research Council framework” has been evaluated. As per MRC framework, a content validity study was performed through an expert panel to assess the scientific and practical appropriateness of the SAS intervention protocol. The content validity index (CVI), including item level-CVI and protocol-level CVI, were calculated to evaluate the consensus level of the expert panel. The true SAS intervention was determined as daily self-administered acupressure on specific acupoints for seven weeks. The placebo SAS was designed as light acupressure on non-acupoints with the same frequency and duration as the valid SAS. Excellent content validity was achieved after one round of expert panel assessment, as all the critical components of the true and placebo SAS protocols were rated as content valid (CVI ranged from 0.86 to 1.00).

The authors planned this study comprehensive literature search including PubMed, Google Scholar, and China National Knowledge Infrastructure (CNKI) were searched using “acupressure”, “acupuncture”, “acupoints stimulation”, “theory”, and “fatigue”, in combination with manual search of books on Traditional Chinese Medicine through the university library. Neurophysiological theories and some TCM theories, particularly the inflammatory theory, yin-yang theory, and zang-fu organs and meridians theory, were identified and utilised to theoretically clarify the potential mechanisms regarding FSDSC in BCS and to inform the selection of appropriate intervention.

The authors wrote MS with recent references, and the methodology sounds good. The article can be acceptable for publication in the journal “International Journal of Environmental Research and Public Health”.

Reviewer 2 Report

Dear Editor and Authors,

Thank you for the opportunity to review the manuscript entitled “An evidence-based somatic acupressure intervention for managing the breast cancer fatigue-sleep disturbance-depression symptom cluster: Development and validation following the Medical Research Council framework”. This study aimed to develop an evidence-based SAS intervention protocol that can be further implemented in a Phase II randomized control trial (RCT) to manage the FSDSC in breast cancer survivors. The Authors concluded that: “A research-informed, theory-driven and practically feasible SAS intervention protocol for the FSDSC management in breast cancer survivors was developed following the MRC framework. The feasibility and acceptability of the SAS intervention will be further tested in breast cancer survivors through a Phase II RCT”. The project itself is very interesting and the reader would like to see the results not only the process of creating the protocol. I am not sure about the policy of the journal and leave it to the Editor’s opinion if the Journal accepts study protocols (“…publishes original articles, critical reviews, research notes, and short communications…”) and if the article structure for original research is appropriate in such case. The content itself indicates a well-designed protocol, which could be published before presenting the results. However, I would suggest:

-          adding some graphic content to highlight the most important aspects of the protocol, e.g., boxes with inclusion/exclusion criteria, planned number of participants, age, sex structure (I have not found these information) and a scheme of research interventions.

-          - adding to the title sth like: “Protocol for the study”.

-          - Is it possible to present a scheme of acupuncture points /figure?/? I am totally unfamiliar with them, and I guess, like most readers…

-         -  Reference style – should be according to Journal’s guidelines and unified

Reviewer 3 Report

great original work

I added minor comments in the manuscript

and I suggest that you make sure to have the latest updates of references because some of them dated from 1986

i realize that some are classic evidence but I suggest to have another look

and make sure to use all relevant and updated evidence.
